# Asymmetric DQN for Partially Observable Reinforcement Learning

**Andrea Baisero**[1]          **Brett Daley**[1]          **Christopher Amato**[1]

[1]Khoury College of Computer Sciences, Northeastern University, Boston, Massachusetts, USA

## Abstract

Offline training in simulated partially observable environments allows reinforcement learning methods to exploit privileged state information through a mechanism known as asymmetry. Such privileged information has the potential to greatly improve the optimal convergence properties, if used appropriately. However, current research in asymmetric reinforcement learning is often heuristic in nature, with few connections to underlying theory or theoretical guarantees, and is primarily tested through empirical evaluation. In this work, we develop the theory of *Asymmetric Policy Iteration*, an exact model-based dynamic programming solution method, and then apply relaxations which eventually result in *Asymmetric DQN*, a model-free deep reinforcement learning algorithm. Our theoretical findings are complemented and validated by empirical experimentation performed in environments which exhibit significant amounts of partial observability, and require both information gathering strategies and memorization.

## 1 INTRODUCTION

Offline training and online execution (OTOE) is a modern reinforcement learning (RL) paradigm in which a learning agent is trained *offline* (i.e., in simulation) before becoming operational *online* (i.e., in the "real" environment). Advantages of OTOE are broad and include safety guarantees, training speed, flexibility, and—the focus of our work—access to privileged information. For all these reasons, OTOE has even become the paradigm of preference in some research cliques, such as that of multi-agent RL, where it is often called centralized training and decentralized execution (CTDE). Privileged information is data which is accessible during offline training, but not during standard on-line training and/or execution. This can take different forms depending on the type of control problem, e.g., other agents' actions and observations in multi-agent RL, or the system's state in partially observable RL (PORL). In OTOE, access to this information is a temporary privilege, available exclusively in the offline phase due to access of the simulation's internal state. However, despite being not available during online execution, such information has the potential (when used appropriately) to improve the agent's overall training performance and/or convergence speed, and therefore its online performance.

In PORL, OTOE and privileged information is most commonly associated with actor-critic methods through a mechanism called *asymmetry*. Asymmetry has a very specific etymological meaning described below; however, we use the term "asymmetry" more loosely to also refer to the general idea of exploiting privileged information during offline training—two concepts which overlap strongly in this work. In actor-critic methods, two separate models are being trained: a *policy* model (representing the agent's behavior) and a *critic* model (representing the agent's evaluation of its situation). Standard actor-critic methods can be said to be implicitly *symmetric* in the sense that both models receive the same information—in PORL, the agent's history. In *asymmetric* actor-critic, this symmetry is broken by providing privileged information to the critic [Pinto et al., 2018, Foerster et al., 2018, Lowe et al., 2017, Yang et al., 2018, Li et al., 2019, Wang et al., 2020, Warrington et al., 2021, Xiao et al., 2021, Baisero and Amato, 2022, Lyu et al., 2022]. This is possible because the critic is exclusively a training construct which is not used or needed during the execution phase. Asymmetry has also been used in some DQN-like RL methods [Rashid et al., 2018, Mahajan et al., 2019, Rashid et al., 2020, Xiao et al., 2020, de Witt et al., 2020], where normally there would not be a secondary model analogous to the critic which is only used during training. In such cases, a second value-based model is introduced exclusively as a means for asymmetry, and which constitutes a training construct analogous to that of the critic in actor-critic.

*Accepted for the 38th Conference on Uncertainty in Artificial Intelligence* (UAI 2022).

However, a substantial majority of prior work in asymmetric RL has proposed heuristic forms of asymmetry primarily verified through empirical evaluations, but which lack the support of a theoretical framework which guarantees the state information is used in an appropriate fashion. We argue that, if unverified by proper theoretical analysis, such methods could quite simply make use of state information in ways which actually hinders the training of a partially observable agent. For example, a well-known result in partially observable control is that the optimal action for a partially observable agent can differ greatly from that of an optimal fully observable agent, and that an optimal partially observable agent might even take actions which an optimal fully observable agent would never take under any circumstance, e.g., information-gathering actions that help the partially observable agent learn something about the environment state, but do not help the fully observable agent.

The ultimate goal of this work is to develop a state-of-the-art asymmetric value-based deep RL algorithm for partially observable control that is supported by a sound theoretical analysis. To reach this goal, we employ a bottom-up approach, focusing first on developing the theory of *asymmetric policy improvement*, i.e., mechanisms through which privileged state information can be integrated into a policy improvement process while retaining optimal convergence guarantees. In practice, we begin by developing *Asymmetric Policy Iteration* (API) and *Asymmetric Action-Value Iteration*, model-based dynamic programming solution methods. We then introduce elements of stochastic training from sample experience which result in *Asymmetric Q-Learning* (AQL), a direct RL successor to API and AAVI. Finally, we introduce value-function approximation which results in *Asymmetric DQN* (ADQN), a method comparable to other state-of-the-art deep RL algorithms, but which is also capable of exploiting state information in a principled fashion. To the best of our knowledge, our work is the first to develop theoretically-driven asymmetric value-based RL.

## 2   RELATED WORK

Privileged information available offline has been used to improve training performances in a wide range of prior single-agent and multi-agent methods which include both policy-based and value-based methods.

In sigle-agent control, Pinto et al. [2018] employ DDPG with an asymmetric state-based critic to handle robot manipulation tasks; belief-grounded networks [Nguyen et al., 2021] uses a belief-based form of asymmetry and a belief-reconstruction task to train the history representation; Warrington et al. [2021], Chen et al. [2020] use imitation learning to train a partially observable agent via a fully observable agent trained offline. Baisero and Amato [2022] show theoretical issues with state-only forms of asymmetry for policy-gradients, and develop a history-state variant.

In multi-agent control, COMA [Foerster et al., 2018] uses a single centralized asymmetric critic which employs the joint observations and/or the environment state. MADDPG [Lowe et al., 2017] and M3DDPG [Li et al., 2019] use multiple centralized asymmetric critics, one for each agent, which employ the joint observations and/or the environment state. R-MADDPG [Wang et al., 2020] uses a recurrent model and a centralized critic which uses the entire histories of all agents; CM3 [Yang et al., 2018] uses a state-only critic for reactive control; MacDec-DDRQN [Xiao et al., 2020] uses a centralized value model to learn individual centralized value models. ROLA [Xiao et al., 2021] uses both centralized and individual asymmetric critics which employ local history and/or state information to estimate individual advantage values. QMIX [Rashid et al., 2018], MAVEN [Mahajan et al., 2019], and WQMIX [Rashid et al., 2020] use a centralized but factored value model to train individual agent value models. Lyu et al. [2022] extend the theory by Baisero and Amato [2022] to the multi-agent case.

## 3   BACKGROUND

In the next subsection, we present some of the background required to understand our work. Section 3.1 formally describes partially observable control problems. Section 3.2 contains a review of non-asymmetric value-based control, in the form of DQN. Section 3.3 covers the definition of *history-state* value functions. In Section 3.4, we present operator notation and useful operators.

### 3.1   POMDPS

A partially observable Markov decision process (POMDP) is a discrete-time control problem represented by tuple $\langle \mathcal{S}, \mathcal{A}, \mathcal{O}, b_0, T, O, R, \gamma \rangle$, where (a) $\mathcal{S}$, $\mathcal{A}$, and $\mathcal{O}$ are state, action, and observation spaces, (b) $b_0 \in \Delta\mathcal{S}$ is an initial state distribution, (c) $T \colon \mathcal{S} \times \mathcal{A} \to \Delta\mathcal{S}$ is a stochastic state transition function, (d) $O \colon \mathcal{S} \times \mathcal{A} \times \mathcal{S} \to \Delta\mathcal{O}$ is a stochastic observation emission function, (e) $R \colon \mathcal{S} \times \mathcal{A} \to \mathbb{R}$ is a reward function, and (f) $\gamma \in [0, 1)$ is a discount factor.

Partially observable control is based on observable *histories*, i.e., the sequences of past actions and observations. The *history* space $\mathcal{H} \doteq (\mathcal{A} \times \mathcal{O})^*$ represents such sequences. To simplify notation, we overload symbol $R$ to also denote the expected reward function on histories $R(h, a) \doteq \mathbb{E}_{s|h}[R(s, a)]$. General partially observable policies take the form of mappings from *histories* to action distributions $\pi \colon \mathcal{H} \to \Delta\mathcal{A}$; however, in this work, we will focus exclusively on *deterministic* policies $\pi \colon \mathcal{H} \to \mathcal{A}$. The goal of the control problem is to find a policy which maximizes the episodic expected *return* $\mathbb{E}\left[\sum_t \gamma^t R(s_t, a_t)\right]$.

Every policy $\pi$ is associated with an action-value function $Q^\pi(h, a)$ which represents the expected return associated

with the agent having observed history $h$, taking action $a$, and then continuing to behave according to the policy $\pi$. $Q^\pi$ is the unique solution to the Bellman equation,

$$Q^\pi(h, a) = R(h, a) + \gamma \, \mathbb{E}_{o|h,a} \left[ Q^\pi(hao, \pi(hao)) \right] . \quad (1)$$

The action-value function associated with the optimal policy $\pi^*$ is denoted as $Q^*$, and is the unique solution to the Bellman *optimality* equation,

$$Q^*(h, a) = R(h, a) + \gamma \, \mathbb{E}_{o|h,a} \left[ \max_{a'} Q^*(hao, a') \right] . \quad (2)$$

**Notation** We use symbols $Q, Q^\pi, Q^*$, and $\hat{Q}$ to denote similar but separate concepts. $Q \colon \mathcal{H} \times \mathcal{A} \to \mathbb{R}$ denotes an arbitrary real-valued function, not necessarily associated with any policy, $Q^\pi$ denotes the value function associated with a policy, $Q^*$ denotes the value function associated with an optimal policy, while $\hat{Q}$ denotes a (deep) parametric model. We denote the space of all such history-action real functions as $\mathcal{Q} \doteq \{ Q \mid Q \colon \mathcal{H} \times \mathcal{A} \to \mathbb{R} \}$. Further, we use $g(Q)$ to denote the policy which acts greedily based on $Q$, i.e., if $\pi = g(Q)$, then $\pi(h) = \text{argmax}_a Q(h, a)$.

## 3.2 DQN

Deep Q-Network (DQN) [Mnih et al., 2015] is a highly successful algorithm for training deep neural networks to control high-dimensional fully-observable Markov decision processes (MDPs) based on reward feedback, and the first to achieve human-level performance on a majority of the Atari 2600 games. Rather than relying on a lookup table to track the estimated expected return for each state-action pair $(s, a)$, DQN learns a parametric function $\hat{Q} \colon \mathcal{S} \times \mathcal{A} \to \mathbb{R}$ to generalize over state-action pairs. The algorithm reformulates the incremental Q-Learning update [Watkins, 1989] as a squared-error minimization problem,

$$\mathcal{L}(s, a, r, s') \doteq \left( r + \gamma \max_{a' \in \mathcal{A}} \hat{Q}(s', a'; \theta^-) - \hat{Q}(s, a; \theta) \right)^2, \quad (3)$$

where $\theta$ is a set of parameters and $\theta^-$ is a time-delayed copy of $\theta$ to stabilize learning. The agent interacts with the environment and stores observed transitions $(s, a, r, s')$ in a replay memory, periodically updating $\theta$ via gradient descent on randomly sampled minibatches of experience [Lin, 1992]. This approach approximates the i.i.d. supervised training setting commonly used for neural networks and required for first-order optimization methods.

**Adapting DQN to Partially Observable Control** The DQN algorithm was primarily designed for fully observable control problems represented as MDPs. Nonetheless, as with many other model-free RL algorithms, generalization to partially observable control is conceptually straightforward, and achievable by replacing state variables with history

variables in the relevant equations, and by employing architectures capable of processing history data. *Frame stacking*, i.e., the practice of concatenating a small number of recent observations, has been found to be sufficient to tackle problems which feature small amounts of partial observability [Mnih et al., 2015]. On the other hand, larger amounts of partial observability generally require longer-term memorization capabilities. For such problems, the standard choice has become that of combining the DQN training algorithm with a recurrent neural network component used to process history data, also known as *Deep Recurrent Q-Network* (DRQN) [Hausknecht and Stone, 2015]. Although some practitioners use the DQN label exclusively to indicate the variant which lacks a recurrent component, our view is that the essence of the DQN algorithm is in its training regime and its losses, rather than the details of which architecture is used. Therefore, in this document, we use the label DQN more broadly to encompass all architectural variants. In practice, because our work focuses on control problems which feature large amounts of partial observability, we employ appropriate algorithmic and modeling choices, i.e., all methods and baselines employ a history-based model $\hat{Q}$, and all models which receive history data employ a recurrent network component to process it.

## 3.3 HISTORY-STATE VALUE FUNCTIONS

Recent theoretical work in asymmetric actor-critic for PORL has employed the notion of a history-state value function $U^\pi(h, s, a)$ [Baisero and Amato, 2022, Lyu et al., 2022], which represents the expected return associated with the agent having observed history $h$, the environment being in state $s$, taking action $a$, and then continuing to behave according to the policy $\pi$. $U^\pi$ is the unique solution to the *history-state* Bellman equation,

$$U^\pi(h, s, a) = R(s, a) + \gamma \, \mathbb{E}_{s', o|s,a} \left[ U^\pi(hao, s', \pi(hao)) \right] . \quad (4)$$

Despite using the state context to represent a more informed measure of the agent's expected return, $U^\pi$ still relates to a partially observable agent which is unable to exploit that privileged information, i.e., the state determines future rewards, observations, and states, but it does not directly determine future actions, which are rather determined indirectly by the history. $U^\pi$ is related to $Q^\pi$ via a simple identity,

$$Q^\pi(h, a) = \mathbb{E}_{s|h} \left[ U^\pi(h, s, a) \right] . \quad (5)$$

We denote the history-state value function associated with the optimal policy as $U^*$. Once again, this notion of optimality is relative to the space of partially observable policies. Among other things, this means that an optimal partially observable policy cannot be recovered by maximizing $U^*$, i.e., generally, there is no guarantee that $\pi^*(h) = \text{argmax}_a U^*(h, s, a)$ for any given value of $s$.

**Notation** We use symbols $U, U^\pi, U^*$, and $\hat{U}$ to denote similar but separate concepts. $U: \mathcal{H} \times \mathcal{S} \times \mathcal{A} \to \mathbb{R}$ denotes an arbitrary real-valued function, not necessarily associated with any policy, $U^\pi$ denotes the value function associated with a policy, $U^*$ denotes the value function associated with an optimal policy, while $\hat{U}$ denotes a (deep) parametric model. We denote the space of all such history-state-action real functions as $\mathcal{U} \doteq \{U \mid U: \mathcal{H} \times \mathcal{S} \times \mathcal{A} \to \mathbb{R}\}$.

### 3.4 OPERATOR NOTATION

To simplify the upcoming math, we make extensive use of operator notation for mappings between $Q$ and $U$ functions.

Operator $B_\pi: \mathcal{Q} \to \mathcal{Q}$ is the Bellman operator defined as $B_\pi Q(h, a) \doteq R(h, a) + \gamma \, \mathbb{E}_{o|h,a} [Q(hao, \pi(hao))]$, with which Equation (1) can be rewritten as $Q^\pi = B_\pi Q^\pi$.

**Lemma 3.1.** *Operator $B_\pi$ is a contraction with fixed point $Q^\pi$ (proof in Appendix A.1.)*

Operator $B: \mathcal{Q} \to \mathcal{Q}$ is the Bellman *optimality* operator defined as $BQ(h, a) \doteq R(h, a) + \gamma \, \mathbb{E}_{o|h,a} [\max_{a'} Q(hao, a')]$ or, equivalently, $B: Q \mapsto B_{g(Q)}Q$, and with which Equation (2) can be rewritten as $Q^* = BQ^*$.

**Lemma 3.2.** *Operator $B$ is a contraction with fixed point $Q^*$ (proof in Appendix A.2.)*

Some operators for $U$ are analogous to those for $Q$. To avoid introducing a separate set of symbols for such cases, we overload the previously defined symbols to include these new meanings; the distinction will remain clear from context, usually as the type of the operator's input/output.

Operator $B_\pi: \mathcal{U} \to \mathcal{U}$ is the Bellman operator defined as $B_\pi U(h, s, a) \doteq R(s, a) + \gamma \, \mathbb{E}_{s',o|s,a} [U(hao, s', \pi(hao))]$, with which Equation (4) can be rewritten as $U^\pi = B_\pi U^\pi$.

**Lemma 3.3.** *Operator $B_\pi$ is a contraction with fixed point $U^\pi$ (proof in Appendix A.3.)*

Operator $E: \mathcal{U} \to \mathcal{Q}$ converts $U$ functions to $Q$ functions by taking the conditional expectation over states, and is defined as $EU(h, a) \doteq \mathbb{E}_{s|h} [U(h, s, a)]$.

**Definition 3.4** (Mutual Consistency). We say that functions $Q$ and $U$ are *mutually consistent* iff $Q = EU$ holds.

## 4 ASYMMETRIC VALUE-BASED PORL

In this section, we present the core of our theoretical and algorithmic contributions, which focus on computing or learning optimal action-values $Q^*(h, a)$ by means of asymmetry. In Section 4.1 we present *Asymmetric Policy Iteration* (API), a solution method for tabular models with optimal

convergence guarantees. In Section 4.2 we present *Asymmetric Action-Value Iteration* (AAVI), an eager variant of API with similar optimal convergence guarantees. In Section 4.3 we relax aspects of AAVI to make it suitable for learning by means of sample experience, and present *Asymmetric Q-Learning* (AQL). In Section 4.4 we introduce value function approximation to improve generalization, and present *Asymmetric DQN* (ADQN), and other related variants.

**Introducing Asymmetry to Value-Based Methods** Two fundamental issues make the use of state information in value-based methods not directly possible: (a) because an action-value model $\hat{Q}(h, a)$ is eventually used for online control, it is constrained by the control problem and cannot directly employ privileged state information; and (b) typical value-based methods do not feature a separate model for the purpose of offline training which may access privileged information (akin to the critic in actor-critic). As such, value-based methods seem fundamentally incompatible with the notion of asymmetry and the use of privileged information. We resolve both issues, and introduce an auxiliary history-state model $\hat{U}(h, s, a)$, trained to model the optimal history-state value function $U^*(h, s, a)$, and used exclusively as a training construct through which to implement asymmetry. Our goal is to train $\hat{U}$ and $\hat{Q}$ jointly so as to converge to the optimal value functions $U^*$ and $Q^*$.

### 4.1 ASYMMETRIC POLICY ITERATION

Consider *Asymmetric Policy Iteration* (API), an iterative process analogous to Policy Iteration [Sutton and Barto, 2018] which employs both history-state and history values to implement asymmetry. API starts from arbitrary initial values and policy $U_0$, $Q_0$, and $\pi_0$, and then uses the following update rules to generate sequences $U_k$, $Q_k$, and $\pi_k$,

$$U_{k+1} \leftarrow \lim_{n \to \infty} B_{\pi_k}^n U_k, \qquad \text{(U-evaluation)} \quad (6)$$

$$Q_{k+1} \leftarrow EU_{k+1}, \qquad \text{(Q-evaluation)} \quad (7)$$

$$\pi_{k+1} \leftarrow g(Q_{k+1}). \qquad \text{(improvement)} \quad (8)$$

The U-evaluation step can be practically implemented as the solution to the system of equations $U_{k+1} = R + \gamma P_{\pi_k} U_{k+1}$, or by using $B_{\pi_k}$ until convergence (see Algorithm 1).

**Theorem 4.1** (API Optimality). *The sequences $U_k$, $Q_k$, and $\pi_k$ generated by API converge to $U^*$, $Q^*$, and $\pi^*$.*

*Proof.* By Lemma 3.1, $U_{k+1}$ equals the fixed point of $B_{\pi_k}$, i.e., $U_{k+1} = U^{\pi_k}$. Then, by Equation (5), $Q_{k+1} = EU^{\pi_k} = Q^{\pi_k}$ and consequently $\pi_{k+1} = g(Q^{\pi_k})$. Therefore, in each iteration and until $\pi^*$ is reached, the next policy $\pi_{k+1}$ is a strict improvement on the previous policy $\pi_k$ (Policy Improvement Theorem, [Sutton and Barto, 2018]). Let $k^*$ be the smallest index such that $\pi_{k^*}$ is optimal; for $k > k^*$, we conclude that $U_k = U^*$ and $Q_k = Q^*$. $\qquad \square$

**Algorithm 1** Asymmetric Policy Iteration (API)

**Require:** $U_0, Q_0, \pi_0$ arbitrarily initialized tabular models.
**Ensure:** $\lim_{k \to \infty}\{U_k, Q_k, \pi_k\} = \{U^*, Q^*, \pi^*\}$.
 1: **for** $k \leftarrow 0, 1, 2, 3, \dots$ **do**
 2:     $U_{k+1} \leftarrow U_k$
 3:     **repeat**
 4:         $U_{k+1} \leftarrow B_{\pi_k} U_{k+1}$
 5:     **until** convergence
 6:     $Q_{k+1} \leftarrow EU_{k+1}$
 7:     $\pi_{k+1} \leftarrow g(Q_{k+1})$
 8: **end for**

---

**Limitations** While API is formally guaranteed to converge optimally, it also has significant practical limitations: (a) API is a solution method which requires a model of the environment, as well as efficient and accurate methods to compute the expectations in the U-evaluation and the Q-evaluation steps. (b) A practical approximation of the limit operator in the U-evaluation step (see Algorithm 1) might itself require multiple iterations to achieve an adequate precision. (c) API requires tabular models $U$ and $Q$, which is not only impractical given that the space of histories grows exponentially with episode lengths, but also makes it not applicable to control problems which have continuous observations or states. (d) Perhaps most importantly, API does not offer any significant advantage compared to its non-asymmetric counterpart PI. Ultimately, both API and PI converge to the same optimal value function $Q^*$; if anything, API requires more memory and computation to achieve the same goal, resulting in a less practical solution method.

**Why API?** In light of the above limitations, particularly the last one, what is then the purpose of API? We argue that API plays two crucial roles: (a) The first is to show that privileged and asymmetric information such as the system state *can* be properly included into a value-based solution process while maintaining formal optimality guarantees. This theoretical aspect is often overlooked in modern asymmetric RL research, which instead tends to focus on heuristic methods and empirical results, and API represents the first theoretical guarantee of this kind for value-based RL. (b) The second is to serve as a basis for other algorithms which do provide practical advantages compared to their non-asymmetric counterparts. Starting from the next subsection, we relax various aspects of API and develop asymmetric value-based algorithms which address each of API's limitations.

## 4.2 ASYMMETRIC ACTION-VALUE ITERATION

The first limitation of API which we address is the presence of the limiting operator in its U-step, which makes practical implementations approximate, and/or inefficient. To this end, consider *Asymmetric Action-Value Iteration* (AAVI),

**Algorithm 2** Asymmetric Action-Value Iteration (AAVI)

**Require:** $U_0, Q_0$ arbitrarily initialized tabular models.
**Ensure:** $\lim_{k \to \infty}\{U_k, Q_k\} = \{U^*, Q^*\}$.
 1: **for** $k \leftarrow 0, 1, 2, 3, \dots$ **do**
 2:     $U_{k+1} \leftarrow B_{g(Q_k)} U_k$
 3:     $Q_{k+1} \leftarrow EU_{k+1}$
 4: **end for**

---

an eager variant of API which uses the following updates,

$$U_{k+1} \leftarrow B_{g(Q_k)} U_k, \qquad \text{(U-evaluation)} \qquad (9)$$
$$Q_{k+1} \leftarrow EU_{k+1}. \qquad \text{(Q-evaluation)} \qquad (10)$$

Compared to API, the improvement step has been folded in the U-evaluation step, removing the need for an explicit policy representation. Further, the U-evaluation step has been simplified to apply operator $B_{g(Q_k)}$ a single time, making for a simple, faster, and more practical implementation (see Algorithm 2) without compromising optimality guarantees. Both aspects make AAVI analogous to Value Iteration [Sutton and Barto, 2018], with the primary differences being the use of action-values and asymmetry.

**Lemma 4.2** (Asymmetric Bellman Equivalence). *For mutually consistent $U$ and $Q$, the identity $EB_{g(Q)}U = BQ$ holds (proof in Appendix A.4.)*

**Theorem 4.3** (AAVI Optimality). *The sequences $U_k$ and $Q_k$ generated by AAVI converge to $U^*$ and $Q^*$.*

*Proof.* We can combine the U-evaluation and Q-evaluation steps, and then use Lemma 4.2 to obtain $Q_{k+1} = EU_{k+1} = EB_{g(Q_k)}U_k = BQ_k$. By induction, $Q_k = B^k Q_0$, which converges to the fixed point of $B$: i.e., $\lim_{k \to \infty} Q_k = Q^*$. This guarantees the existence of some iteration $k^*$ such that $g(Q_k) = \pi^*, \forall k \geq k^*$. Therefore, $U_k = B_{\pi^*}^{k-k^*} U_{k^*}, \forall k \geq k^*$, and $U_k$ converges to the fixed point of $B_{\pi^*}$: i.e., $\lim_{k \to \infty} U_k = U^*$. ◻

## 4.3 ASYMMETRIC Q-LEARNING

Like all dynamic programming methods, API and AAVI make extensive and often unrealistic assumptions like the model of the environment and being able to compute exact expectations. To bypass many of these requirements, we can employ incremental stochastic updates based on sequentially sampled transitions. We call this new method *Asymmetric Q-Learning* (AQL), as it generalizes the iterative Q-Learning algorithm [Watkins, 1989] to asymmetric PORL.

To handle the randomness induced by the sample transitions, the algorithm must average over the samples using a variable stepsize parameter $\alpha_k \in [0, 1]$. At each iteration $k$, the agent

**Algorithm 3** Asymmetric Q-Learning (AQL)

**Require:** $U, Q$ mutually consistent tabular models.
**Ensure:** $\{U, Q\} \rightarrow \{U^*, Q^*\}$.
1: **while** True **do**
2:   Initialize history and state $(h, s)$
3:   **while** $s$ is not terminal **do**
4:     Choose action $a$ from $\epsilon$-greedy policy on $Q$
5:     Take action $a$, observe $r, s', o$
6:     $y \leftarrow r + \gamma U(hao, s', \text{argmax}_{a'} Q(hao, a'))$
7:     $U(h, s, a) \leftarrow (1 - \alpha)U(h, s, a) + \alpha y$
8:     $Q(h, a) \leftarrow (1 - \alpha)Q(h, a) + \alpha y$
9:     $(s, h) \leftarrow (s', hao)$
10:   **end while**
11: **end while**

**Algorithm 4** Asymmetric DQN (ADQN)

**Require:** $\hat{U}, \hat{Q}$ deep models parameterized by $\theta$.
1: Initialize parameters $\theta$
2: Initialize and prepopulate episode buffer
3: **while** True **do**
4:   From the simulated environment, sample episodes and append them to the episode buffer
5:   From the episode buffer, sample batch of transitions $\{(h_i, s_i, a_i, r_i, s'_i, o_i)\}_{i=1}^{N}$
6:   $L_U \leftarrow \frac{1}{N} \sum_{i=1}^{N} \mathcal{L}_{\hat{U}}(h_i, s_i, a_i, r_i, s'_i, o_i).$
7:   $L_Q \leftarrow \frac{1}{N} \sum_{i=1}^{N} \mathcal{L}_{\hat{Q}}(h_i, s_i, a_i, r_i, s'_i, o_i).$
8:   Perform a gradient step on $\theta$ using $\nabla_\theta(L_U + L_Q)$
9: **end while**

samples a transition $(h_k, s_k, a_k, r_k, s_{k+1}, o_k)$ and conducts AAVI-like updates[1] on the respective entries of $U_k$ and $Q_k$.

For notational brevity, we first define the following targets:

$$Y_k(h, s, a) \doteq \begin{cases} \left(B_{g(Q_k)}U_k + w_k\right)(h, s, a) & \text{for } (h_k, s_k, a_k) \\ U_k(h, s, a) & \text{otherwise} \end{cases}$$ (11)

$$Z_k(h, a) \doteq \begin{cases} \left(EB_{g(Q_k)}U_k + v_k\right)(h, a) & \text{for } (h_k, a_k) \\ Q_k(h, a) & \text{otherwise} \end{cases}.$$ (12)

Here, $w_k \in \mathcal{U}$ and $v_k \in \mathcal{Q}$ are zero-mean noise processes that represent the randomness in the environment and action selection at iteration $k$. AQL then conducts the following updates based on the stochastic targets $Y_k$ and $Z_k$:

$$U_{k+1} \leftarrow U_k + \alpha_k(Y_k - U_k),$$ (13)
$$Q_{k+1} \leftarrow Q_k + \alpha_k(Z_k - Q_k)\Pr(s_k \mid h_k).$$ (14)

Note that the targets $Y_k$ and $Z_k$ are defined elementwise such that only one entry of $U_k$ and $Q_k$—the one associated with $(h_k, s_k, a_k)$—is updated for any given index $k$. When the stepsizes $\alpha_k$ are annealed towards zero at an appropriate rate, AQL converges optimally despite the noisy updates.

**Theorem 4.4** (AQL Optimality). *Assume stepsizes $\alpha_k$ satisfying the following asymptotic conditions,*

$$\sum_{k=0}^{\infty} \alpha_k = \infty, \qquad \sum_{k=0}^{\infty} \alpha_k^2 < \infty.$$ (15)

*If $Q_0, U_0$ are mutually consistent ($Q_0 = EU_0$), then the sequences $Q_k$ and $U_k$ generated by AQL converge to $Q^*$ and $U^*$ with probability 1 (proof in Appendix A.5.)*

---

[1]The Q-evaluation step of AAVI (Equation (10)) can equivalently be expressed as $Q_{k+1} \leftarrow EB_{g(Q_k)}U_k$, which is the form AQL employs to guarantee optimal convergence.

Factor $\Pr(s_k \mid h_k)$ is necessary to ensure that $U_k$ and $Q_k$ remain mutually consistent throughout the process; a necessary condition for optimal convergence. $\Pr(s_k \mid h_k)$ can be interpreted as a scaling factor which makes $U_k$ and $Q_k$ update at relatively comparable rates. For any given "full" update on $U_k(h_k, s_k, a_k)$ the corresponding update on $Q_k(h_k, a_k)$ should be scaled down to a "partial amount" relative to the likelihood of $s_k$. While we were able to remove other forms of model-based requirements, $\Pr(s_k \mid h_k)$ remains, leaving AQL just shy from reaching both optimal convergence and concrete practicality at the same time. While it may be possible to approximate this factor in other model-free ways, AQL remains primarily a conceptual algorithm also due to the requirement of a tabular model over histories. Either way, AQL serves as a fundamental basis to derive the next iteration of asymmetric value-based RL.

### 4.4 ASYMMETRIC DQN

When acting in POMDPs with high-dimensional observations and states, a tabular-lookup method such as AQL becomes infeasible. In such instances, we must introduce function approximation to generalize over similar experiences. The use of approximation sacrifices the optimal convergence guarantee established by Theorem 4.4, but is necessary to scale algorithms to significantly more challenging partially observable environments. Nevertheless, the value function approximations are an orthogonal matter to how privileged state information is used, and we expect the sound theoretical principles upon which AQL is built will help asymmetric deep methods even when relying on function approximation.

Our primary algorithmic contribution here is *Asymmetric DQN* (ADQN), an asymmetric variant of DQN derived by introducing value function approximation to AQL. We first replace the tabular-lookup models $U$ and $Q$ of AQL with parametric differentiable models $\hat{U}$ and $\hat{Q}$. In practice, these are implemented as deep neural networks whose architectures are chosen according to the structure of the states and observations emitted by the POMDP.

To facilitate the substitution, we must reformulate the stochastic update rules of AQL as squared-error loss minimization. For the rest of the section, due to spacing concerns, we will use $\hat{\pi} = g(\hat{Q})$ as a shorthand to represent actions selected greedily on $\hat{Q}$, i.e., $\hat{\pi}(h) = \text{argmax}_a \hat{Q}(h, a)$. Given a single environment interaction $(h, s, a, r, s', o)$, the corresponding losses can be defined as

$$\mathcal{L}_{\hat{U}} = (r + \gamma \,\text{SG}\,[\hat{U}(hao, s', \hat{\pi}(hao))] - \hat{U}(h, s, a))^2 \,, \tag{16}$$

$$\mathcal{L}_{\hat{Q}} = (r + \gamma \,\text{SG}\,[\hat{U}(hao, s', \hat{\pi}(hao))] - \hat{Q}(h, a))^2 \,, \tag{17}$$

where SG is the *stop-gradient* function which indicates that gradient calculation should not consider the enclosed terms. It is worth noting that $\mathcal{L}_{\hat{U}}$ and $\mathcal{L}_{\hat{Q}}$ use the same target to train $\hat{U}$ and $\hat{Q}$. The crucial difference is that $\hat{U}$ is in able to model the target as a function of $s$, while $\hat{Q}$ is unable to do so, and can at only model the expectation of the target over values of $s$. In a way, these losses approximately enforce a "loose" form of mutual consistency $\hat{Q} \approx E\hat{U}$. In practice, the term is generated by "target networks" that rely on stale parameters to stabilize learning when bootstrapping [Mnih et al., 2015]; the stale parameters are periodically updated by copying the main parameters. The total loss $\mathcal{L}_{\hat{U}} + \mathcal{L}_{\hat{Q}}$ can be jointly minimized with respect to the parameters by a single backpropagation step, efficiently approximating the interleaved updates of AQL.

When the function approximators are nonlinear (as is often the case for neural networks), training will fail if the gradient updates are conducted on sequentially collected transitions that are not i.i.d. [Mnih et al., 2015]. The second critical modification to AQL is therefore the adoption of experience replay [Lin, 1992] in order to decorrelate training experiences. Rather than training on a sample immediately when it is collected, each POMDP transition $(h, s, a, r, s', o)$ is deferred to a first-in first-out replay memory. Periodically, when it is time to train the networks, a minibatch of several experiences is sampled from the replay memory; gradients for these samples are computed and averaged together to estimate the true gradient of the joint loss $\mathcal{L}_{\hat{U}} + \mathcal{L}_{\hat{Q}}$, which in turn is used to improve the parameters.

**Why ADQN?** Having finally addressed the practical disadvantages of API, it is worthwhile to reconsider the "why" question again, this time focusing on why one would prefer to use ADQN compared to DQN. Ultimately, the purpose of both algorithms is to train an approximate $\hat{Q} \approx Q^*$ through which optimal control can be executed, and both algorithms should *in theory* converge to very similar approximations. What is then the advantage of ADQN over DQN? Similarly to the asymmetric actor-critic case [Baisero and Amato, 2022], the advantage is a practical one associated with the difficulties of learning an appropriate representation of history $\phi(h)$, which is one of the major bottlenecks in PORL.

History representations are sequence models which notoriously requires lots of data and processing power for proper training. To further compound on this issue, the quality of the data used to train the history representation in PORL is directly related to the quality of $\hat{Q}(\phi(h), \cdot, \cdot)$, which in turn depends on the quality of the history representation itself; unsurprisingly, it can be quite hard to bootstrap the training of an improved history representation when starting from a poor history representation. Note, however, that learning an appropriate state representation $\phi(s)$ is much simpler than learning $\phi(h)$ due to the non-sequential nature of individual states, i.e., the $\phi(s)$ representation model has fixed input and output sizes, and can generally be modeled using a simpler feed-forward architecture. In ADQN, the issues associated with learning a proper history representation are alleviated by the fact that its training is bootstrapped not only on the history representation itself, but also on the state representation. Even when the history representation is poor, we can expect the state representation to contain sufficient contextual information to allow $\hat{U}(\phi(h), \phi(s), \cdot)$ to model meaningful values, which in turns helps further bootstrap the learning of the history representation $\phi(h)$, the history model $\hat{Q}(\phi(h), \cdot)$, and the respective implicit policy $g(\hat{Q})$.

Next, we consider some variants of interest of ADQN.

### 4.4.1 Variance-Reduced ADQN

In this variant, we approximate the target of $\mathcal{L}_{\hat{Q}}$ from Equation (17) as $r + \gamma \hat{U}(hao, s', \hat{\pi}(hao)) \approx \hat{U}(h, s, a)$, which holds particularly well once $\hat{U}$ has been trained sufficiently. Therefore, this variant uses the following losses,

$$\mathcal{L}_{\hat{U}} = (r + \gamma \,\text{SG}\,[\hat{U}(hao, s', \hat{\pi}(hao))] - \hat{U}(h, s, a))^2 \,, \tag{18}$$

$$\mathcal{L}_{\hat{Q}} = (\text{SG}\,[\hat{U}(h, s, a)] - \hat{Q}(h, a))^2 \,. \tag{19}$$

This approximate target results in lower variance throughout the entire training process at the cost of introducing bias primarily in the early stages of training; a trade-off which may result advantageous in some control problems.

### 4.4.2 State-Only ADQN

Some prior work in asymmetric RL has adopted heuristic forms of asymmetry which uses state-only (i.e., history-less) value functions $U(s, a)$. Such form of asymmetry is however associated with fundamental theoretical issues which may severely compromise the learning performance, ranging from potentially being ill-defined, to introducing bias into the learning process [Baisero and Amato, 2022]. Such issues are inherently related to partial observability, and their effects scale with the amount of partial observability that the agent is subject to, as well as the agent's "reactiveness", i.e. the amount of history that is willfully ignored by the agent

itself to select actions. Nonetheless, this state-only form of value function may be more useful than others in control problems which have small amounts of partial observability, such as vision-based tasks with an occlusion-free view of the environment [Pinto et al., 2018, Baisero and Amato, 2022]. Although our main focus is control problems with significant amounts of partial observability, we are still interested in formulating a state-only variant of ADQN as an additional baseline, and as reference for future work which may focus on the kinds of control problems where it thrives.

In this state-only variant, we redefine the parametric model $\hat{U}(s, a)$ to ignore history, and adopt the following losses,

$$\mathcal{L}_{\hat{U}} = (r + \gamma \operatorname{SG}[\hat{U}(s', \hat{\pi}(hao))] - \hat{U}(s, a))^2 , \quad (20)$$

$$\mathcal{L}_{\hat{Q}} = (r + \gamma \operatorname{SG}[\hat{U}(s', \hat{\pi}(hao))] - \hat{Q}(h, a))^2 . \quad (21)$$

### 4.4.3 Reduced-Variance State-Only ADQN

This variant applies a state-only variant of the variance reduction approximation from Section 4.4.1 to state-only ADQN. In this case, we approximate the target of $\mathcal{L}_{\hat{Q}}$ from Equation (21) as $r + \gamma \hat{U}(s', \hat{\pi}(hao)) \approx \hat{U}(s, a)$,

$$\mathcal{L}_{\hat{U}} = (r + \gamma \operatorname{SG}[\hat{U}(s', \pi(hao))] - \hat{U}(s, a))^2 , \quad (22)$$

$$\mathcal{L}_{\hat{Q}} = (\operatorname{SG}[\hat{U}(s, a)] - \hat{Q}(h, a))^2 . \quad (23)$$

## 5 EVALUATION

We perform an empirical evaluation of our proposed ADQN method and its variants in a variety of environments which feature significant amounts of partial observability.

**Methods** We compare the performances of 5 value-based PORL algorithms, denoted as follows:

- **DQN** is the standard non-asymmetric DQN algorithm;

- **ADQN** and **ADQN-VR** are the history-state ADQN algorithms from Sections 4.4 and 4.4.1; and

- **ADQN-State** and **ADQN-State-VR** are the state-only ADQN algorithms from Sections 4.4.2 and 4.4.3.

**Environments** Evaluations are run on 5 partially observable navigation tasks which require information gathering strategies and memorization of the past:

- **Heaven-Hell-3** and **Heaven-Hell-4** [Bonet, 1998], corridor environments where the agent must reach the exit to *heaven* and avoid the exit to *hell*, but must first backtrack to visit a *priest* to learn which exit is which;

- **Car-Flag** [Nguyen, 2021], a 1-dimensional continuous control variant of **Heaven-Hell**;

- **Cleaner** [Jiang and Amato, 2021], a maze environment where two agents must reach all tiles to clean them. In our experiments, the two agents are treated as a single agent, and controlled in a centralized fashion; and

- **GV-MemoryFourRooms-7x7** [Baisero and Katt, 2021], a dynamically generated gridworld with 4 connected rooms, where the agent must reach the *good* exit and avoid the *bad* exit, but must first find and memorize a *beacon* to learn which is which.

A more thorough description of these environments can be found in Appendix C of Baisero and Amato [2022].

Each method is trained and evaluated using code available as a public repository[2]. For each environment and algorithm, we perform an independent grid-search over some hyper-parameters of interest (see Appendix D), and select the combination of hyper-parameters which results in the best final performance and learning stability (prioritizing final performance if necessary). To improve the statistical significance of the results, each combination of environment, algorithm, and hyper-parameters is run 20 independent times.

### 5.1 RESULTS AND DISCUSSION

Figure 1 shows the results of these evaluations, which broadly confirm our theoretical analysis on asymmetric value-based PORL, the practical advantage of employing history-state forms of evaluation to aid partially observable control, and confirm the superiority of ADQN compared to other similar symmetric and asymmetric variants. This evaluation further confirms other recently developed theoretical analysis on asymmetric PORL, i.e., that state-only forms of asymmetry are inadequate to handle non-trivial amounts of partial observability [Baisero and Amato, 2022].

Across the board, **ADQN** and **ADQN-VR** outperform all baselines in final performance, convergence speed, and/or overall learning stability. The contrast between methods is particularly stark in Figures 1a and 1b, where **ADQN** and **ADQN-VR** are not just the only methods that demonstrate any substantial improvement, but are also able to reach optimal performance. On the other hand, the state-only variants fail to outperform even the **DQN** baseline in most environments (with a single exception discussed later), which further confirms the theoretical issues that have been recently associated with state-only forms of asymmetry. Broadly, the variance-reduced variants **ADQN-VR** and **ADQN-State-VR** only differ in relatively minor ways from their default counterparts. Such differences can be found in Figures 1a and 1d, where **ADQN** is more stable or has better convergence properties than **ADQN-State-VR**, and in Figures 1c and 1d, where **ADQN-State-VR** has better final convergence values than **ADQN-State**. This seems to indicate

---

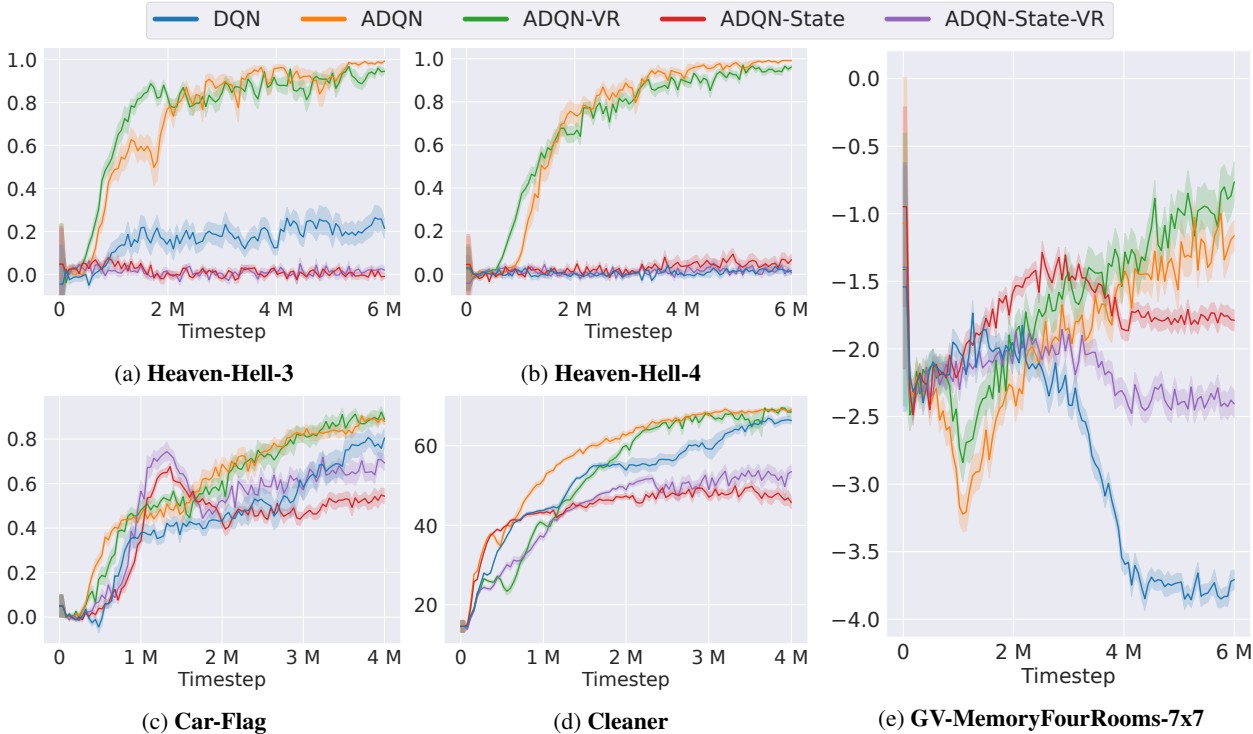

(a) **Heaven-Hell-3**  (b) **Heaven-Hell-4**  (c) **Car-Flag**  (d) **Cleaner**  (e) **GV-MemoryFourRooms-7x7**

Figure 1: Performance curves showing episodic returns averaged over the last 100 completed episodes, with statistics computed over 20 independent runs. The shaded areas represent one standard error around the mean.

that the type of asymmetry (history-state or state-only) is a larger contributor to overall performance than the choice of using the standard or the variance-reduced variant of the same method. In Figure 1e too, **ADQN** and **ADQN-VR** outperform all other baselines. However, these results also represent an interesting exception to some of the above analysis, e.g., **ADQN-State** outperforms **DQN** and **ADQN-VR** outperforms **ADQN**. To explain this, we note that this is the only task not reliably solved by any of the methods, which is likely due to the highly dynamic nature of the randomly generated map and object locations. In fact, the fact that some of the trends found in the other results do not also appear here may be explained by the fact that none of the methods achieve their full potential performance.

## 6  CONCLUSIONS

OTOE is a RL framework where agents are trained offline in a simulated environment, which allows temporary access to privileged information which would otherwise be unavailable, like the partially observable environment's state. Asymmetry is a common mechanism through which such privileged information can be used during training, and has the potential to greatly boost learning performance and efficiency when implemented correctly. However, modern work in asymmetric RL tends to focus on unproven heuristics which lack a theoretical justification. In this work, we filled

this void and developed the theory of asymmetric value-based RL. We achieved our primary goal of developing a theoretically-sound asymmetric value-based RL algorithm by employing a bottom-up approach, and by first focusing on the base theory of asymmetric policy improvement. This took the form of API, a conceptual solution method with strict optimal convergence guarantees but concrete practical limitations. Then, we applied a series of relaxations to API which addressed those limitations and ultimately resulted in ADQN, a practical and competitive deep RL algorithm. We performed an empirical evaluation to compare the performances of ADQN and its variants to standard non-asymmetric DQN in a series of environments which are specifically selected to exhibit high levels of partial observability, and which require information-gathering strategies and memorization of the past. In all these environments, ADQN achieved the best performance, even solving control problems which standard DQN could not. Overall, our evaluation confirmed the potential offered by privileged information, the importance of using it in principled and theoretically-guided ways, and the overall success our ADQN algorithm in partially observable control problems. Future work may focus on extending ADQN to the multi-agent control case, which poses further learning challenges, on finding applications where state-only ADQN may thrive (such as vision-based robotic tasks with little partial observability), and on extending the evaluation of ADQN in more complicated partially observable vision-based tasks.

## Author Contributions

Andrea Baisero conceived the idea, developed proofs, ran experiments, and wrote the paper. Brett Daley developed proofs and wrote the paper. Christopher Amato supervised.

## Acknowledgements

This research was funded by NSF award 1816382.

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
