# OpenReview forum: "Asymmetric DQN for Partially Observable Reinforcement Learning"
_auai.org/UAI/2022/Conference — UAI 2022 Poster_

### Official Review · Reviewer_ph4L · 2022-04-11

**Q2(1) Originality/Novelty:** 3
**Q2(2) Significance/Impact:** 3
**Q2(3) Correctness/Technical Quality:** 3
**Q2(6) Clarity Of Writing:** 3
**Q6 Overall Score:** 6
**Q8 Confidence In Your Score:** 2

**Q1 Summary And Contributions:**

A theoretically-sound asymmetric value-based RL algorithm is proposed in this paper, where the base theory of asymmetric policy improvement (API) is firstly developed, which has strict optimal convergence guarantees but concrete practical limitations. Then a series of relaxations are applied to API to address those limitations.

**Q2 Assessment Of The Paper:**

More detailed information regarding each of these aspects is given below:

**Q2(4) Quality Of Experiments (Optional):**

2: Fair: The experimental evaluation is weak: important baselines are missing, or the results do not adequately support the main claims.

**Q2(5) Reproducibility:**

2: Fair: Key resources (e.g., proofs, code, data) are unavailable but key details (e.g., proof sketches, experimental setup) are sufficiently well-described for an expert to confidently reproduce the main results.

**Q3 Main Strengths:**

The paper is clearly written and a theoretically-driven asymmetric value-based RL algorithm is proposed. Also, the paper is well-organized, that a bottom-up approach is employed by focusing first on developing the theory of asymmetric policy improvement, and then a series of relaxations are applied to finally introduce the value-function approximation-based approach, namely ADQN.

**Q4 Main Weakness:**

To me, the main concern of the paper is the experiments, where only four toy problems were used for the evaluation. Moreover, only DQN was used for the comparison. As DQN is not designed for the partially observable domains and as in the literature, many other techniques have been proposed to address the partially observable issue, the effectiveness of the proposed method is unclear.

**Q5 Detailed Comments To The Authors:**

In this paper, a theoretically-sound asymmetric value-based RL algorithm is proposed, where the base theory of asymmetric policy improvement (API) is firstly developed, which has strict optimal convergence guarantees but concrete practical limitations. Then a series of relaxations are applied to API to address those limitations.

In general, the paper is clearly written and a theoretically-driven asymmetric value-based RL algorithm is proposed. Also, the performances of the proposed techniques are verified in several environments with significant amount of partial observability.

However, as can be seen from the evaluation, the performances of ADQN and its variants are only compared with DQN, while DQN is not designed for the partially observable problems, and it is unclear what is the input for the DQN algorithm. Actually, in the DRL literature, many techniques have been proposed for addressing the partially observable issue, e.g., DRQN, the performances of such methods should be compared and discussed. Moreover, only four toy problems were used for the evaluation, and the scalability of the proposed approach is also unclear.


**Q7 Justification For Your Score:**

In this paper, a theoretically-driven asymmetric value-based RL algorithm is proposed, the paper is well written and well organized. The main weakness of the paper is that only four toy problems were used for the evaluation, and the scalability of the proposed approach is unclear. Moreover, as only DQN was used for the comparison, the effectiveness of the proposed approach is also unknown.

**Q9 Complying With Reviewing Instructions:**

1: Yes.

---

### Official Review · Reviewer_CCvT · 2022-04-12

**Q2(1) Originality/Novelty:** 3
**Q2(2) Significance/Impact:** 2
**Q2(3) Correctness/Technical Quality:** 3
**Q2(6) Clarity Of Writing:** 3
**Q6 Overall Score:** 6
**Q8 Confidence In Your Score:** 2

**Q1 Summary And Contributions:**

To fill the gap that existing asymmetric RL methods lack theoretical guarantees, the authors first presented the optimality theory of Asymmetric Policy Iteration, and to relax the assumptions of API, they further developed Asymmetric Action-Value Iteration, and Asymmetric Q-Learning (AQL). Finally, they introduced value function approximation to derive practical algorithms, asymmetric DQN (ADQN), including its three variants.

**Q2 Assessment Of The Paper:**

More detailed information regarding each of these aspects is given below:

**Q2(4) Quality Of Experiments (Optional):**

2: Fair: The experimental evaluation is weak: important baselines are missing, or the results do not adequately support the main claims.

**Q2(5) Reproducibility:**

2: Fair: Key resources (e.g., proofs, code, data) are unavailable but key details (e.g., proof sketches, experimental setup) are sufficiently well-described for an expert to confidently reproduce the main results.

**Q3 Main Strengths:**

This paper is well written and organized with clear motivations. The authors have made non-trivial advances in the field of asymmetric value-based RL where current research lacks theoretical optimality guarantees but has only empirical verification. Their proposed theories on API, AVVI, and AQL look interesting and complemented.

**Q4 Main Weakness:**

I think it is hard to say that the experiments are sufficient to validate the proposed theoretical findings or verify the efficacy of the proposed methods. The reasons are as follows. Firstly, the authors only took the traditional method DQN as a non-asymmetric comparison method, while all the other 4 methods are the proposed ones, which looks less persuasive about the advantages brought from the asymmetry mechanism. It might be better to take more non-asymmetric baselines or explain the reasons why not compare other RL methods. Secondly, there exist other asymmetric RL methods, as shown in Section 2. It might be better to take other asymmetry methods into comparison to verify the efficacy of the proposed methods, e.g., A2C-asym-s in Baisero and Amato [2022].




**Q5 Detailed Comments To The Authors:**

Please see Q3 and A4.

0. The authors used ADQN and its variants in experiments. Since the use of approximation sacrifices the optimal convergence guarantee in ADQN and its variants, why could the authors say “these evaluations largely confirm the theoretical analysis from our work” on Page 8? To confirm the theory, it is straightforward to consider API, AAVI, and AQL in experiments, but the authors use ADQN for confirmation. Please explain it.

1. It is not clear what privileged information means in the paper. We know that such information is the potential to improve the training performances in RL, however, what it actually is or stands for is unclear.

2. There are some descriptions that are unclear to me:
- In Algorithms 1 (line 2) and 2 (line 2), how could obtain the operators $B_{\pi_k}$ and $B_{g(Q_k)}$ to update the $U_{k+1}$? Are they served as some kind of prior information?
-  In Page 7, how could we obtain Eqs. (16) and (17) from Eq.(15) using the target approximation?


**Q7 Justification For Your Score:**

Please see Q3-Q5.

**Q9 Complying With Reviewing Instructions:**

1: Yes.

---

### Official Review · Reviewer_A8Gu · 2022-04-15

**Q2(1) Originality/Novelty:** 2
**Q2(2) Significance/Impact:** 2
**Q2(3) Correctness/Technical Quality:** 2
**Q2(6) Clarity Of Writing:** 3
**Q6 Overall Score:** 4
**Q8 Confidence In Your Score:** 4

**Q1 Summary And Contributions:**

This paper proposed a history-based DQN algorithm to solve the partially observable reinforcement learning (PORL) problem, by introducing a history-state-based value function U into the Q value function update, named asymmetric DQN(ADQN). Also, the authors proposed a state-only ADQN version, ADQN-state, to solve the RL problems with small amounts of partial observability. Some empirical studies showed the superiority of ADQN compared with vanilla DQN and ADQN-state.

**Q2 Assessment Of The Paper:**

More detailed information regarding each of these aspects is given below:

**Q2(4) Quality Of Experiments (Optional):**

1: Poor: The experimental evaluation is flawed or the results fail to adequately support the main claims.

**Q2(5) Reproducibility:**

3: Good: Key resources (e.g., proofs, code, data) are available and key details (e.g., proofs, experimental setup) are sufficiently well-described for competent researchers to confidently reproduce the main results.

**Q3 Main Strengths:**

1. Good writing and presentation;
2. Well-defined technologies;
3. Convincing math.

**Q4 Main Weakness:**

1. Lacks discussions on previous related works;
2. Some conclusions lack supports;
3. Empirical studies are too simple.

**Q5 Detailed Comments To The Authors:**

1. This paper proposed a Q value version of the asymmetric RL method in [1]. Usually, asymmetric information is embodied in the privileged information of the critic (value function estimator) compared with the actor (policy). But how could this privileged information be embodied in a Q-valued-based algorithm? The authors might argue that the U function contains more privileged information compared with the Q function. However, what is the difference between this method and the previous DQN-based POMDP algorithms? In view of this paper, history usage as asymmetric information seems no different from the previous recurrent neural network-based algorithms. In addition, the empirical results showed that without history, the U function would hurt the performance of DQN, since ADQN-state performs worse than vanilla DQN.

2. Some usages or conclusions lack the discussion of why these could happen. For example, in equations 14 and 15, why use a stop-gradient function on the U function? In the second paragraph of page 7 ("Why ADQN?" part), "learning an appropriate state representation $\phi(s)$ is much simpler due to the non-sequential nature of states.". How could states contain the "non-sequential nature"?

3. The experiments only contain the comparations of the performance, without any empirical investigation on whether the usage of the U function is reasonable for the asymmetric information. Meanwhile, the only contender is vanilla DQN, which is too weak, since the vanilla DQN is not designed for the POMDP problem even. In addition, ADQN-state performs worse than DQN, which seems the U function here is superfluous without the history information.

**Q7 Justification For Your Score:**

As discussed above, the lack of discussions with previous related works, vague support for the conclusions, and insufficient empirical studies degenerate the contribution of this paper.

**Q9 Complying With Reviewing Instructions:**

1: Yes.

---

### Decision · Program_Chairs · 2022-05-15

**Decision:**

Accept (Poster)

**Comment:**

Meta Review: The paper describes an asymmetric version of DQN for partially observable environments.  This is very interesting work that shows how to use state information at training time while ensuring that states remain hidden at execution time.  The paper provides very interesting insights and advances the state of the art for partially observable RL.  The authors are encouraged to follow the suggestions of the reviewers while preparing the final version of their paper.